# The Effect of Surface Modification of Gold Nanotriangles for Surface-Enhanced Raman Scattering Performance

**DOI:** 10.3390/nano10112187

**Published:** 2020-11-02

**Authors:** Joachim Koetz

**Affiliations:** Institute for Chemistry, University of Potsdam, Karl-Liebknecht-Strasse 24-25, Haus 25, 14476 Potsdam, Germany; koetz@uni-potsdam.de; Tel.: +49-331-977-5220

**Keywords:** undulated, spiked and crumble gold nanotriangles, SERS enhancement factor, dimerization of 4-nitrothiophenol, AOT bilayer, PEI coating

## Abstract

A surface modification of ultraflat gold nanotriangles (AuNTs) with different shaped nanoparticles is of special relevance for surface-enhanced Raman scattering (SERS) and the photo-catalytic activity of plasmonic substrates. Therefore, different approaches are used to verify the flat platelet morphology of the AuNTs by oriented overgrowth with metal nanoparticles. The most important part for the morphological transformation of the AuNTs is the coating layer, containing surfactants or polymers. By using well established AuNTs stabilized by a dioctyl sodium sulfosuccinate (AOT) bilayer, different strategies of surface modification with noble metal nanoparticles are possible. On the one hand undulated superstructures were synthesized by in situ growth of hemispherical gold nanoparticles in the polyethyleneimine (PEI)-coated AOT bilayer of the AuNTs. On the other hand spiked AuNTs were obtained by a direct reduction of Au^3+^ ions in the AOT double layer in presence of silver ions and ascorbic acid as reducing agent. Additionally, crumble topping of the smooth AuNTs can be realized after an exchange of the AOT bilayer by hyaluronic acid, followed by a silver-ion mediated reduction with ascorbic acid. Furthermore, a decoration with silver nanoparticles after coating the AOT bilayer with the cationic surfactant benzylhexadecyldimethylammonium chloride (BDAC) can be realized. In that case the ultraviolet (UV)-absorption of the undulated Au@Ag nanoplatelets can be tuned depending on the degree of decoration with silver nanoparticles. Comparing the Raman scattering data for the plasmon driven dimerization of 4-nitrothiophenol (4-NTP) to 4,4′-dimercaptoazobenzene (DMAB) one can conclude that the most important effect of surface modification with a 75 times higher enhancement factor in SERS experiments becomes available by decoration with gold spikes.

## 1. Introduction

The first synthesis of gold nanoparticles dates back to the 5th or 4th century for making ruby glass [1]. Later, alchemists tried to use red-colored, transparent gold solution as an elixir of life. However, the everlasting youth concept failed, but up until now colloidal gold sols have been used in different biomedical applications, especially in the field of cancer therapy and early diagnosis [2]. Nowadays it is well known that the red color of 10 and 50 nm-sized spherical gold nanoparticles with an ultraviolet (UV) absorption at about 520 nm is related to the coherent oscillation occurring at the surface electrons of metal nanoparticles, i.e., the localized surface plasmon resonance (LSPR) [3,4]. Plasmonics, the science and application of noble metal structures interacting with light, has received significant attention in the last few decades [5,6].

Using asymmetric nanoparticles, like gold nanorods or gold nanotriangles (AuNTs), the absorption maximum can be shifted into the near-infrared (NIR) wavelength region between 800–1300 nm (so called biological window), which is of special relevance for biomedical imaging, where tissue is relatively transparent [7]. Plasmonic nanoparticles with intense localized electric fields can be successfully used for detecting and enhancing weak processes via surface-enhanced Raman scattering (SERS). Recently, an extended review (from more than 50 authors) with a reference list of 959 papers about present and future of SERS was published [8]. SERS enables the detection of analyte molecules deposited on plasmonic nanoparticles. However, the sensitivity of the experiments strongly depends on the shape and size of the nanoparticles used.

Nanorods and nanotriangles are two classes of anisotropic nanoparticle with excellent SERS properties due to the large electromagnetic fields at the ends of nanorods [9,10] or vertices of nanotriangles [11,12]. For AuNTs, the sharpness of the edges and tips is of special relevance for improving the enhancement factor (EF) and, therefore, the application in catalysis [13,14]. Furthermore, large-scale self-assembly of the AuNTs on a substrate is of high importance for SERS performance [13,15,16].

However, there is still an open question: how to increase further the EF of a given asymmetric gold nanoparticle system?

One way to improve the EF of plasmonic nanoparticles is to modify the platelet surface by decorating it with nanospheres or nanospikes [17]. Upon excitation with light large electromagnetic fields can be localized between the nanoparticles fixed at the particle surface, and Raman-active molecules, sandwiched between spike tips, can intensify SERS signals [18]. Therefore, the arrangement and sharpness of the tips is of special relevance for tuning the plasmon resonance [19,20]. Using the antenna effect of spikes located on the surface of gold nanorods SERS signals can be intensified, significantly [21]. High-density islands with small nanogaps on the surface of nanotriangles lead to excellent SERS performance with an EF up to 2.3 × 10^5^, one magnitude higher in comparison to smooth AuNTs [22]. The main goal of this short review is to evaluate the effect of morphological transformations on the nanoplatelete surface on the enhancement effect in SERS experiments.

## 2. Smooth (Bare) Dioctyl Sodium Sulfosuccinate (AOT)-Coated Gold Nanotriangles (AuNTs)

Well-shaped AuNTs can be synthesized by a silver-free growth of cetyl trimethylammonium chloride (CTAC)-coated seeds in the presence of iodide anions [13,23] or a seedless synthesis through oxidative etching [24]. However, the asymmetric growth of gold nanorods or gold nanotriangles strongly depends on the type of surfactant and the corresponding counterion used. Impurities of the surfactant component either induces or disrupts the asymmetric growth [25].

Our own strategy was focused on the formation of triangular gold nanoparticles in a multivesicular template phase [11,26]. Small angle X-ray scattering (SAXS) measurements show that the gold nanoplatelets with a mean platelet thickness of 7.5 ± 1 nm (in full agreement with transmission electron microscopy (TEM) experiments) are formed via an Ostwald ripening growth mechanism [27]. Hereby, a fraction of very stable ultrathin nanoplatelets can be separated by a depletion flocculation in the presence of dioctyl sodium sulfosuccinate (AOT) micelles. After washing and centrifugation at 13,000 rpm, TEM micrographs of the stock solution indicate that 79% of the ultrathin nanoplatelets are nanotriangles with an edge length of 175 ± 17 nm [11]. Therefore, in the following part the nanoplatelets are called AuNTs. High-resolution transmission electron microscopy (HRTEM) investigations demonstrate that the AuNTs are composed of {111} facets on the top and bottom surface [11]. These ultrathin smooth AuNTs can be deposed on a substrate in large scale by a spontaneous film casting after adding an ethanol-toluene mixture to the aqueous, green-colored AuNT solution [15]. Figure 1 shows SEM micrographs of a well-ordered monolayer of AuNTs on a silicon wafer. The negative zeta potential of −75 mV, obtained by electrophoretic light scattering [27], indicates the presence of an AOT-bilayer at the {111} smooth platelet surface, which is in good agreement with additionally performed MD simulations [28].

To verify the photocatalytic activity of the AuNT monolayer a model reaction for the plasmon driven dimerization of 4-nitrothiophenol (4-NTP) to 4,4′-dimercaptoazobenzene (DMAB) was investigated [11,15]. Therefore, the AuNTs were assembled by a 4-NTP monolayer via S–Au bonding. The characteristic Raman peaks of 4-NTP at 1077 cm^−1^ for C–H bending, at 1327–1335 cm^−1^ for NO_2_ symmetric stretching, and at 1575 cm^−1^ for C=C stretching modes were intensified, which means the plasmonic AuNT monolayer enhances the signal intensity up to a factor of EF = 2.76 × 10^4^ [11,15], somewhat higher than cetyl trimethylammonium bromide (CTAB)-based nanoprisms with an EF of 1.6 × 10^4^ [16]. The substrate enhancement factor of the densely packed AOT-stabilized AuNT monolayer covered with 4-NTP is already a remarkably high signal yield in comparison to the relevant literature [13,14]. Scarabelli et al. found in colloidal AuNT dispersions an EF of 1.2 × 10^5^ for benzenethiol [13] and Kuttner et al. an EF of 5.6 × 10^4^ for mercaptobenzoic acid [14].

Hereby, the EF is defined as the ratio of SERS intensity on the SERS substrate (I_SERS_) to Raman intensity of solid 4-NTP molecules (I_solid_) at the strongest vibration (symmetric stretching of the NO_2_ groups) in relation to the number of adsorbed (N_adsorbed_) and solid (N_solid_) 4-NTP molecules:EF = I_SERS_ × N_solid_/I_solid_ × N_adsorbed_(1)

(N_adsorbed_ = 5.65 × 10^6^; N_solid_ = 1.39 × 10^11^)

Furthermore, a dimerization to DMAB can be observed at a laser power ≥5 mW, proved by three new peaks at 1134, 1387 and 1434 cm^−1^ assigned to C–N symmetric stretching and N=N stretching vibrational modes of DMAB (compare Figure 2).

## 3. Surface Modification of Smooth AOT-Coated AuNTs

In the following part, different strategies for an oriented overgrowth with gold and silver nanoparticles were pursued. Note that morphologic transformations strongly depend on the type of coating shell. Therefore, we have reloaded the AOT shell by adding the oppositely charged polycation poly(ethyleneimine) (PEI) and the oppositely charged cationic surfactant benzylhexadecyldimethylammonium chloride (BDAC). Furthermore, we have replaced the AOT bilayer with the negatively charged biopolymer sodium hyaluronate (HA) in comparison to a direct gold nanoparticle synthesis in the AOT bilayer with ascorbic acid as reducing agent in the presence of silver salts (AgNO_3_) (compare Scheme 1).

### 3.1. Undulated AuNTs after Gold Nanoparticle Formation in a Polyethyleneimine (PEI) Shell

It is already well established that the cationic PEI can be successfully used as a reducing and stabilizing agent for the gold nanoparticle formation [29,30,31]. Time dependent ultraviolet–visible (UV-vis) spectroscopic and SAXS experiments in an aqueous diluted PEI solution at 100 °C have shown that in the first time period (up to 5 min), 2 nm-sized gold clusters are formed, indicated by an absorption maximum at 350 nm [31]. These clusters grow up to 5 nm-sized nanoparticles with an UV absorption maximum at about 520 nm in the following 20 min [31]. Consequently, electrosteric stabilized gold nanoparticles become available [29]. Therefore, we have reloaded the smooth AOT-stabilized AuNTs by a PEI layer and have synthesized gold nanoparticles in the PEI layer, surrounding the AuNTs, schematically shown in Scheme 2.

Surprisingly, TEM micrographs show an undulated morphology with half spheres on the single-crystal surface at top and bottom and the edges, as to be seen in Figure 3. Based on corresponding SAXS measurements the following mechanism is discussed: in a first step a slow penetration of Au^3+^ ions through the PEI-AOT shell leads to the formation of gold clusters. The preliminarily formed 2 nm-sized Au clusters crystallize onto the {111} facets on the platelet surface and grow up to individual half spheres [31].

These undulated AuNTs were separated two times by centrifugation for 8 min at 13,000 rpm, before deposition on a silicon wafer via film casting after adding an ethanol-toluene mixture, analogous to the preparation procedure for the smooth AuNTs. SEM micrographs show a close-packed layer of undulated AuNTs on a silicon wafer after evaporation of the liquid [31]. SERS experiments with 4-NTP (performed according to the protocol of smooth AuNTs) indicate an increased photocatalytic activity of the undulated AuNTs. Note, that the close-packed undulated AuNT layer is not a perfect monolayer. Therefore, the enhancement factor was calculated assuming two monolayers of 4-NTP, adsorbed on both sides of the platelets. The resulting enhancement factor with EF = 6 × 10^4^ (N_adsorbed_ = 8 × 10^6^; N_solid_ = 1 × 10^11^) [31], is still two times higher than the EF of smooth AuNTs.

### 3.2. Crumble AuNTs after Replacing the AOT Bilayer by Hyaluronic Acid

By adding the negatively charged biopolymer HA, to the negatively charged AOT-based AuNTs, the negative zeta potential is decreased from −59 mV to −33 mV, which can be related to the substitution of the surfactant bilayer by the polymer HA (compare Scheme 3) [32].

After removing the excess HA and AOT molecules by centrifugation at 13,000 rpm, the resulting biocompatible smooth AuNTs can be used for further surface modification experiments. By adding a gold chloride precursor solution in the presence of ascorbic acid (AA) as reducing agent and silver salts as shape-directing components to the AuNT solution, one can observe a gold nanoparticle formation process in the HA shell on the surface of the AuNTs. The reduction process of the Au^3+^ ions, penetrated to the oppositely charged HA layer, with AA molecules being very fast and only controlled by the amount of Au^3+^ ions in solution. At a low amount of HAuCL_4_ in solution, individual spherical gold nanoparticles are formed at the platelet surface. By increasing amount of HAuCL_4,_ the individual gold nanoparticles welded partially into each other and crumble islands were formed, as seen in Figure 4 [32].

SAXS measurements show that the thickness of the crumble AuNTs is increased up to 14 nm [32]. SERS spectra of 4-NTP molecules adsorbed on the smooth (bare) AuNTs in comparison to the crumble AuNTs demonstrate the strong enhancement effect due to the decoration with crumbles on the platelet surface (compare Figure 5) [32]. The corresponding EF = 5.6 × 10^5^ was calculated one order higher in comparison to the smooth (bare) AuNTs [32].

### 3.3. Spiked AuNTs Formed in the AOT Bilayer

It is already well established that gold nanostars with sharp tips and spikes are of special relevance for SERS experiments [18,19,20,33,34,35,36,37]. Gold nanostars can be synthesized by a simple one-pot procedure or a seed-mediated synthesis in presence of different capping agents, e.g., the cationic surfactants CTAB or CTAC [38], the anionic surfactants sodium dodecyl sulfate (SDS) or AOT [36,39], the polymers polyvinyl pyrrolidone (PVP) or poly diallyldimethylammonium chloride (PDADMAC) [33,34,40].

Our own idea was to synthesize sharp spikes in the AOT shell surrounding the smooth AuNTs under similar conditions, i.e., by reducing Au^3+^ ions in the AOT shell after adding ascorbic acid as reducing component in the presence of Ag^+^ ions as spike-directing agent. The concept is shown in Scheme 4.

In a first step, the Au^3+^ and Ag^+^ ions turn to the negatively charged AOT bilayer, before ascorbic acid molecules attach to the bilayer surface, visualized by corresponding MD simulations [41]. Therefore, the reduction process of Au^3+^ ions is realized in the AOT shell under similar experimental conditions, used for the gold nanostar synthesis [39]. TEM micrographs of the resulting AuNTs show that spikes are formed at the platelet surface as well as the edges of the AuNTs (compare TEM micrograph in Figure 6). EDX measurements reveal that the spikes contain both noble metals, i.e., gold (80%) and silver (20%). The Au spikes are surrounded by a silver layer [41] in similarity to nanostars synthesized under similar conditions in a concentrated AOT solution [39].

The extraordinary effect of the spikes is shown in the Raman intensities of the spiked (modified) AuNTs in comparison to the smooth (bare) AuNTs. The calculated excellent enhancement effect with EF = 8 × 10^5^, is about 75 times higher after surface decoration with tips and spikes.

### 3.4. Silver-Decorated AuNTs Formed in an AOT/Benzylhexadecyldimethylammonium Chloride (BDAC) Shell

Kuttner et al. have reported on the seedless synthesis of AuNTs using 3-butenoic acid in the presence of the cationic surfactant BDAC as capping agent [14], and Tebbe et al. have shown a time-dependent silver-overgrowth in BDAC bilayers attached to the surface of gold nanorods [10]. These experiments demonstrate special features of the cationic surfactant BDAC with regard to the asymmetric gold nanoparticle formation and surface modification in the presence of silver salts.

By adding BDAC to our AOT-stabilized AuNTs one can observe a reloading of the nanoparticles due to a coating with a BDAC bilayer, experimentally shown by a change of the negative zeta potential from −21 mV to +41 mV [42]. Additional performed MD simulations show strong electrostatic interactions between AOT and BDAC molecules, resulting in AOT/BDAC mixed catanionic micelles, which strongly adsorb on the {111} facets of the gold surface [43].

These AOT/BDAC-coated AuNTs can be successful used as a template for making silver nanoparticles in the AOT/BDAC shell as to be seen in Scheme 5.

The catanionic AOT/BDAC layer hinders the fast transport of the Ag^+^ ions leading to the formation of individual Ag nanoparticles on the AuNT surface (compare Figure 7) [42], in contrast to a complete silver-overgrowth, observed in presence of a BDAC bilayer in the absence of AOT [10].

The advantage of the Au@Ag AuNTs decorated with Ag nanoparticles is the ability to tune the LSPR absorption between 1300 and 800 nm depending on the AgNO_3_ concentration. Therefore, the absorption maximum can be shifted to the wavelength of the excitation laser of the Raman microscope at 785 nm, which is of high relevance for SERS performance.

Figure 8 shows the effect of decoration with silver nanoparticles on the UV-vis and SERS spectra of rhodamine RG6 molecules assembled on the Au@Ag AuNTs deposited on glass. The preparation procedure was in accordance with the former discussed protocol for smooth AuNTs. The resulting EF = 5.1 × 10^5^ is in a similar high order, already observed for crumble AuNTs [32].

## 4. Conclusions

Laser excitation at 785 nm generates a highly localized field at the tips and gaps of ultra-flat gold nanotriangles as well as hot electrons and plasmonic heat, which is of special relevance for the plasmon-driven photodimerization of 4-NTP [44]. Consequently, spectroscopic signals of molecules adsorbed at the particle surface, e.g., dyes like 4-NTP or rhodamine RG6, are strongly enhanced, experimentally shown in SERS experiments, and documented in the corresponding enhancement factor EF. However, a direct comparison between EF values of different research groups is complicated, because of morphological differences between the nanoparticles and various SERS experiments (in solution or on different substrates). Therefore, we have focused the discussion on our own experiments based on the bare (smooth) AuNT system with an already high enhancement factor of 2.76 × 10^4^.

A novelty of the AOT-stabilized AuNTs is their long-time stability and the possibility of surface modification in the AOT-bilayer. Ultrafast X-ray diffraction measurements have shown that AOT-stabilized AuNTs stay intact at a base temperature of 24 K with a pump wavelength of 400 nm and a fluence of 2.9 mJ/cm^2^ [45]. Ultrafast X-ray diffraction induces a primary oscillatory motion with a single damped out-of-plane breathing mode with a period of 3.6 ps [45]. Furthermore, we have shown that the microscopic distribution of heat dictated by the spot size of the light focus plays a crucial role in the design of the AOT-stabilized plasmonic nanoreactor [46]. Simultaneous measurements of the Stokes and anti-Stokes regions of the SERS spectrum enables in-situ temperature measurements. The results indicate that hot electrons emitted from the tips trigger the dimerization process from 4-NTP to DMAB [44]. Toxicological investigations of AOT- and polymer-coated AuNTs have shown potential applications for imaging and hyperthermia [47]. In particular, less toxic heparin-coated AuNTs with the highest nontoxic concentration of 50 µg/mL show a promising cellular uptake up to 70% [47].

SERS performance of the modified AOT-stabilized AuNT platelets with different-shaped gold and silver nanoparticles under similar experimental conditions show an increase of the EF:by a **factor of 2 by a decoration with gold half spheres** after reloading the AOT shell with PEI [31].by a **factor of 20 by a decoration with gold crumbles** after replacing the AOT shell by a hyaluronate shell [32].by a **factor of 20 by a decoration with silver nanoparticles** after reloading the AOT shell with BDAC [42].by a **factor of 75 by a decoration with gold spikes** after a direct formation in the AOT shell [41].

Firstly, the results show that the decoration with spherical gold nanoparticles can improve the SERS performance, significantly.

Secondly, the arrangement of the gold nanoparticles in the form of crumbles or islands on the platelet surface can show an extraordinary effect with an enhancement factor one magnitude higher. Similar excellent results can be obtained by a crystallization of silver nanoparticles on the platelet surface, due to a shift of the UV absorption to the wavelength of the excitation laser of the Raman microscope.

Thirdly, sharp gold spikes covered by metallic silver give the best results with the highest EF about two magnitudes higher.

These statements seem to be of general relevance for the improvement of the photo-catalytic activity of plasmonic substrates used in SERS experiments. However, for each noble metal system used as plasmonic substrate, the controlled nanoparticle formation in the surfactant or polymer shell is of highest relevance and should be optimized.

## 5. Experimental Section

To ensure the reproducibility all experiments were performed according to the following protocol for the AuNT monolayer formation and SERS performance with 4-NTP.

### 5.1. Self-Assembled AuNT Monolayer Formation

One droplet of the AuNT dispersion was dropped on a silicon wafer. Through the injection of 25 µL of an ethanol-toluene (5:1) mixture, the AuNTs were transferred to the air–liquid interface, and after solvent evaporation a self-assembled AuNT monolayer film was formed on the substrate, without observing a coffee-ring effect.

### 5.2. Surface-Enhanced Raman Scattering (SERS) Performance

The silicon wafer with the AuNT monolayer was immersed in 10 mM ethanolic solution of 4-NTP for 6 h to ensure a complete binding via the thiol groups. The substrate was washed several times in water and ethanol for a complete removal of excess molecules. The wafer was dried in air before being used in the confocal Raman microscope with a laser excitation at 785 nm. The laser power was varied between 1 mW and 10 mW. To ensure the reproducibility and to compare the results with each other, Raman spectra of solid 4-NTP and 4-NTP adsorbed on bare AuNTs were performed additionally.

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
