# Peer review of "The Effect of Surface Modification of Gold Nanotriangles for Surface-Enhanced Raman Scattering Performance"

_nanomaterials, 2020, doi:10.3390/nano10112187_

Round 1

Reviewer 1 Report

The author well addressed all my comments, so I recommend publication as it is.

Reviewer 2 Report

Requested changes have been made. The paper can be published in the present form.

This manuscript is a resubmission of an earlier submission. The following is a list of the peer review reports and author responses from that submission.

Round 1

Reviewer 1 Report

Accept in present form.

Reviewer 2 Report

The manuscript is declared as a review but it is not clear to me what is the main goal of this review. The author is using mostly his 4 previous papers in the review....reviewing SERS performance of Au nanotriangles decorated with Ag nanoparticles which didn't show a very strong SERS effect (EF such as 104-105 especially for R6G dye is not impressive). Moreover, there is nothing about the reproducibility of the SERS measurement, which is very important for the routine application of any new SERS nanostructural surfaces. In conclusion, I don't recommend this paper be published as a review in Nanomaterials. 

Reviewer 3 Report

I felt the review “The Effect of Surface Modification of Gold Nanotriangles for Surface-Enhanced Raman Scattering Performance” by J. Koetz interesting and complete. Due to morphological differences between the nanoparticles and various SERS experiments, I found a clever decision to focus the paper on its proper nanostructured materials and experiments, however without loosing other important references.

Therefore, I suggest to publish the paper after minor revisions, as listed below.

When you say “an UV absorption at about 520 nm” and similar espressions as at line 22, 37, 41-42, 126-127, I wonder how green light could be considered UV….it may be better change UV in UV-visible!

Line 45 and 122 “succesful” may be better use “succesfully”.

line 56-59  “However, there is still an open question: How to increase furthermore the EF of a given asymmetric gold nanoparticle system?

One way to improve the catalytic activity of plasmonic nanoparticles is to modify the platelet surface by decorating”…

I was wondering about the relationship between EF increase and catalytic activity of NP. May be that the sentences could be better rephrased.